# Strain rate of stretch affects crossbridge detachment during relaxation of intact cardiac trabeculae

**Bertrand C. W. Tanner[1], Bradley M. Palmer[2], Charles S. Chung[3]***

**1** Department of Integrative Physiology and Neuroscience, Washington State University, Pullman, Washington, United States of America, **2** Department of Molecular Physiology and Biophysics, University of Vermont, Burlington, Vermont, United States of America, **3** Department of Physiology, Wayne State University, Detroit, Michigan, United States of America

* cchung@med.wayne.edu

## Abstract

Mechanical Control of Relaxation refers to the dependence of myocardial relaxation on the strain rate just prior to relaxation, but the mechanisms of enhanced relaxation are not well characterized. This study aimed to characterize how crossbridge kinetics varied with strain rate and time-to-stretch as the myocardium relaxed in early diastole. Ramp-stretches of varying rates (amplitude = 1% muscle length) were applied to intact rat cardiac trabeculae following a load-clamp at 50% of the maximal developed twitch force, which provides a first-order estimate of ejection and coupling to an afterload. The resultant stress-response was calculated as the difference between the time-dependent stress profile between load-clamped twitches with and without a ramp-stretch. The stress-response exhibited features of the step-stretch response of activated, permeabilized myocardium, such as distortion-dependent peak stress, rapid force decay related to crossbridge detachment, and stress recovery related to crossbridge recruitment. The peak stress was strain rate dependent, but the minimum stress and the time-to-minimum stress values were not. The initial rapid change in the stress-response indicates enhanced crossbridge detachment at higher strain rates during relaxation in intact cardiac trabeculae. Physiologic considerations, such as time-varying calcium, are discussed as potential limitations to fitting these data with traditional distortion-recruitment models of crossbridge activity.

## Introduction

Changes in strain and strain rate are well-appreciated modifiers of muscle contractility. For example, muscle length changes before or during activation in permeabilized and intact cardiac muscle preparations are often used to determine force redevelopment, force enhancement, preload dependence, and more. In contrast, the relationship between muscle strain, strain rate, and relaxation is less well understood. Prior studies using isolated, permeabilized myofibrils have shown that stretching reduces the duration of the slow phase of relaxation, thereby causing the rapid-relaxation phase to start earlier [1]. Length vibrations have also been

**Data Availability Statement:** Source Data for Figs 2, 3 and 5 are available at doi:10.6084/m9.figshare.24796689.

**Funding:** Funding provided by the National Institutes of Health National Heart, Lung, and Blood

Institute (R01HL149164 (BCWT), R44HL137603 (BMP), and R01HL151738 (CSC)), the American Heart Association (23TPA1074093 https://doi.org/10.58275/AHA.23TPA1074093.pc.gr.172307 (BCWT) and 18TPA34170169 (CSC)), and National Science Foundation (2312925 (BCWT) and 1660908 (BMP)). The funders had no role in study design, data collection and analysis, decision to publish, or preparation of the manuscript.

**Competing interests:** The authors have declared that no competing interests exist.

shown to accelerate relaxation in an otherwise isometric twitching, intact trabecula [2]. Recently, we have shown that the rate of lengthening is a key and sufficient modifier of myocardial relaxation, wherein Mechanical Control of Relaxation was not due to afterload or tension. Rather, Mechanical Control of Relaxation is most dependent upon strain rate of a stretch just prior to the start of diastole (start of relaxation) in an intact, electrically stimulated rat trabecula [3]. In this prior study, the strain rate was modulated by maintaining a constant afterload, which meant that faster strain rates occurred at timepoints later into diastole than the slower strain rates. The current study was designed to control the strain rate and time-to-stretch; thus, eliminating a potential confounding variable among the different strain rates. The current study also builds off prior work focusing on crossbridge detachment by utilizing ramp-stretch protocols.

## Methods

### Intact cardiac trabecula mechanics

Cardiac trabeculae were obtained from 7 female Sprague Dawley rats (Charles River Laboratory, 4.38±0.39 months) during studies approved by the Institutional Animal Care and Use Committee at Wayne State University. Briefly, rats were exsanguinated under isoflurane anesthesia. Hearts were rapidly transferred into a cold Perfusion Solution [in mM: 113 NaCl, 4.7 KCl, 0.6 $KH_2PO_4$, 1.2 $MgSO_4$, 12 $NaHCO_3$, 10 $KHCO_3$, 10 2-[4-2-hydroxyethyl)piperazin-1-yl]ethanesulfonic acid (HEPES), 30 Taurine, 5.5 glucose, 10 2,3-butanedione monoxine (BDM)] and cannulated via the aorta. The heart was then perfused with up to 5 mL of Perfusion Solution to minimize clotting and then pinned to a Sylgard 184 (Dow Chemical, Midland MI)-coated dish for dissection. Six (6) trabeculae were obtained from the right ventricle where free-standing trabeculae are more likely to be present, and one (1) from the left ventricle when it was available. A trabecula was transferred to a stage (802D-160, Aurora Scientific, Aurora ON) filled with Tyrodes Solution [in mM: 140 NaCl, 5.4 KCl, 1.8 $CaCl_2$, 1 $MgCl_2$, 10 HEPES, 10 glucose] and mounted between a force transducer and motor (403A and 322, Aurora Scientific). The muscle was paced at 0.5 Hz at 25°C, allowed to twitch isometrically, and equilibrate for at least 30 minutes. The muscle length (pin-pin) and cross-sectional area (CSA) were measured at the optimal length (length resulting in maximal developed force, Lo).

After equilibration, isometric twitching was maintained except for a single twitch during each acquisition. For the experimental procedure, we first measured a normal twitch followed by a reference load-clamp twitch by controlling the developed force signal using a custom controller interfacing in SLControl data acquisition software [3–5]. The clamp was typically held at 50% of the developed, peak systolic force from the prior "normal" twitch, unless otherwise stated. The clamp was maintained until minimum length was achieved; the muscle relaxed isometrically and preload was restored prior to the subsequent beat (red traces, Fig 1A). After at least 6 seconds of isometric twitches, a twitch with a ramp stretch was acquired by repeating the load-clamp with the inclusion of a ramp stretch (blue traces, Fig 1A). Ramp stretches ended at various times following the end of load clamp (= time-to-stretch) throughout relaxation, starting 10 ms after the end of the clamp. Ramp stretches occurred at strain rates of 25, 100, 250, and 1000 s$^{-1}$, muscle lengths/second. The stretch amplitude was 1% of the muscle (pin-to-pin) length.

### Data analysis

Data was analyzed using a custom Matlab (Mathworks, Natick MA) script to determine parameters. Force was normalized to stress for each muscle via dividing force by the CSA. Strain was calculated by dividing the change in muscle length by the length. The start of the

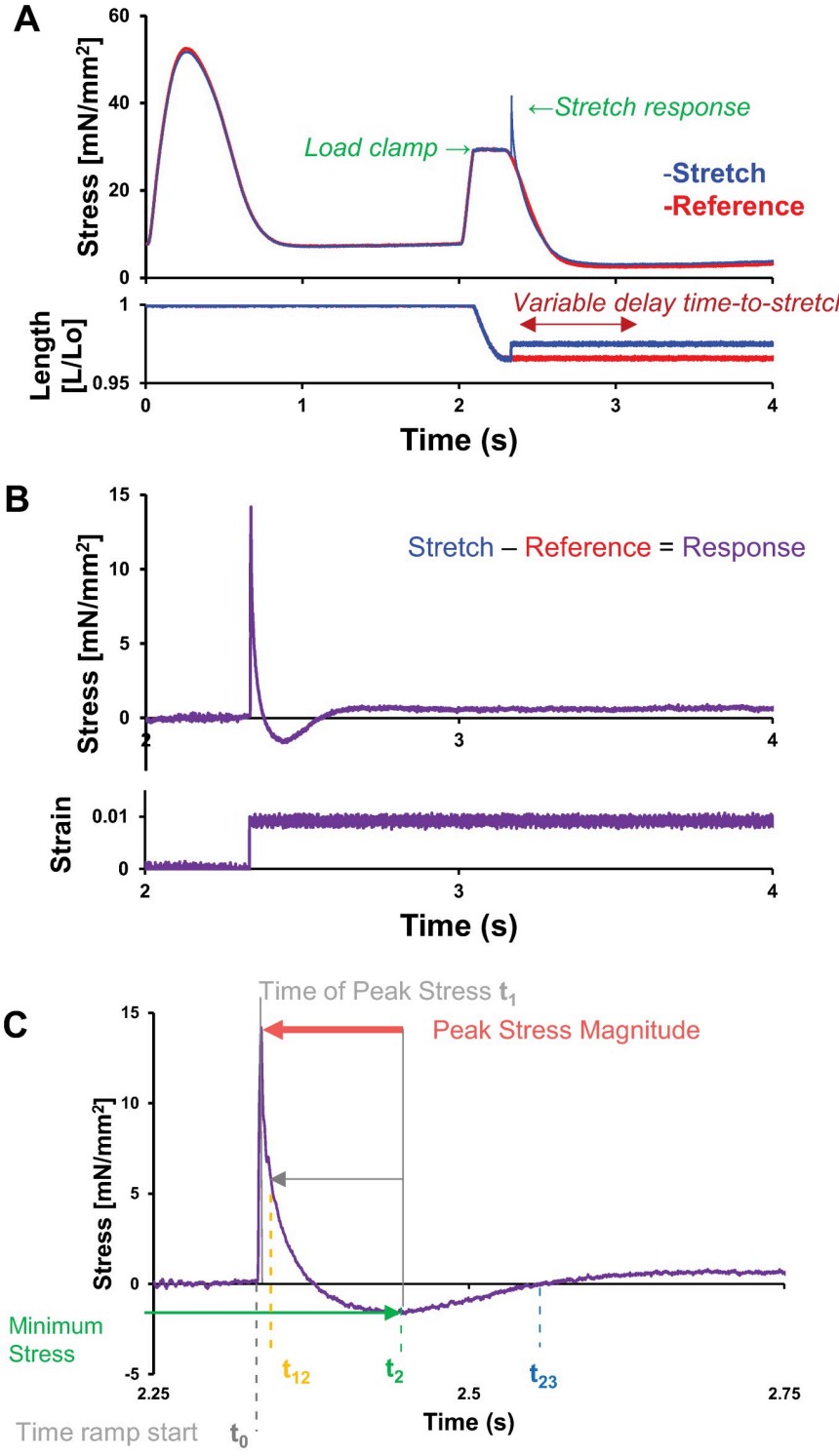

**Fig 1. Example experimental protocol for the twitch, load-clamp, and ramp-stretch following a load-clamp.** A) Example data traces for the measured stress and muscle length are plotted against time to illustrate the paired twitch and load-clamp protocols. Each twitch was followed by a load-clamp to maintain 50% of maximal twitch force, following which the trabecula was either i) held at isometric length as the muscle relaxed (= reference in red) or ii) ramp-stretched at 1% muscle length and four strain rates (25–1000 s⁻¹) then held at isometric length as the muscle relaxed (= stretch in blue). B) The resulting data for stress and strain vs. time (= response in purple) were calculated as the difference between the reference and ramp-stretch traces in panel A. C) Replotting the stress vs. time response

from panel B (upper) across a condensed time duration highlights the more dynamic portions of the response to stretch. Important time-points and stress values that we extracted from these traces are listed in panel C. Data shown for a ramp-stretch strain rate of 1000 $s^{-1}$.

stretch was determined from the strain data, where initial time = 0 sec. at $t_0$, which was used for the model fits to the modulus response (described below). Timepoints for the peak ($t_1$), minimum ($t_2$), the half time of the stress decay ($t_{12}$), and the half time of stress recovery ($t_{23}$) were determined from the stress data (Fig 1). Time values are reported in reference to $t_1$ when reporting the $t_{12}$, $t_2$, and $t_{23}$ values. The timing values $t_2$, or $t_{23}$ were excluded from analysis when they occurred more than 250 ms after $t_1$, as they were not properly detected (i.e. no nadir existed).

### Statistical analysis

Statistical analyses of factors were performed in SPSS 27 (IBM, Armonk, NY) using a Generalized Linear Mixed Model. Each trabecula was treated as a subject and the model included a full factorial analysis of the time after end clamp and strain rate. Pairwise contrasts were performed using a sequential Bonferroni post-hoc test for main factors and $\alpha = 0.05$ was set a priori.

## Results

### Effect of stress on the peak stretch response

Ramp-stretches were performed at a variety of amplitudes and strain rates, at times dictated by the end of the load clamp to the peak of the stretch (time-to-stretch). Four traces are shown in Fig 2A, which illustrate increasing values of time-to-stretch (i.e. the duration between end of load clamp and beginning of stretch). Peak-stress decreased as time-to-stretch increased, shown for a set of stretches in one trabecula using one strain rate (Fig 2B). This indicates that the stretch response scales with the developed stress during relaxation. Normalizing the peak stress-responses to the peak stress value measured 20 ms after the end of the load clamp (because the time-to-stretch value of 20 ms was available for all muscles), values were independent of strain rate (Fig 2C). The normalized peak stress-response also scaled with the reference stress just before the ramp-stretch began (Fig 2D), and was independent of the strain rate. These data indicate that the peak stress scales with the time-to-stretch and the number of bound crossbridges, which declines as relaxation progresses.

### Effects of strain rate and time-to-stretch on the magnitudes of the stress-response

Without any normalization, the peak stress-response decreased as time-to-stretch increased ($p < 0.001$), while the peak stretch response increased as strain rate increased ($p = 0.004$; Fig 3A). As may be expected from the peak stress data presented in Fig 2, this response depends upon the number of bound crossbridges, which decreases throughout diastole. The slowest strain rate (25 $s^{-1}$) was most different compared to the other strain rates that were tested; this slowest strain rate produced the smallest peak stress-response. These strain rate data suggest that the elastic response of the crossbridges, contributing to peak stress, increased as crossbridges were strained at increasingly faster stretches. Note that the diastolic plateau of the reference twitch occurs near 300 ms after the end of the load clamp.

The minimum stress-response (nadir, stress at $t_2$) increased with time-to-stretch ($p < 0.001$), but was not dependent on strain rate (Fig 3B). The minimum stress-response became more

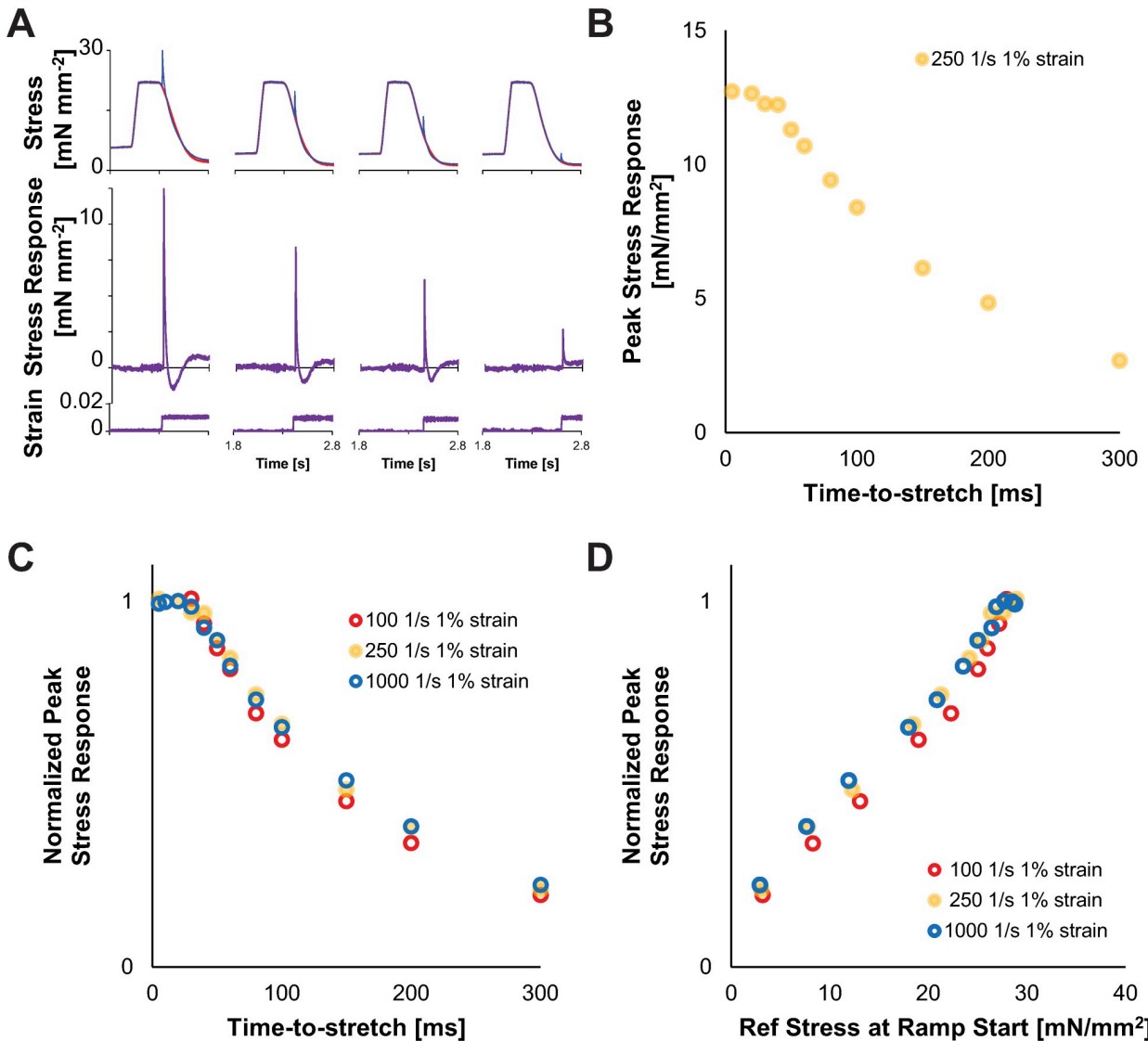

**Fig 2. Peak stress decreases following a ramp-stretch as the time of stretch (post-load clamp) increases, consistent with the twitch relaxation throughout diastole.** A) Example reference (red) and stretched (blue) traces are plotted against time for absolute stress (upper) and the calculated stress-response (middle) and strain (lower). Note that the duration between the end of the clamp and the beginning of the stretch, which defines "time-to-stretch" increases from left to right for the examples in panel A. Traces from experiments at a strain of 1% ML, time-to-stretches of 20, 100, 150 and 300 ms, and strain rate of 250 $s^{-1}$ within a single trabecula. B) An example of peak stress decreasing from multiple time points as time-to-stretch increased; same trabecula as in panel A. C) Normalized (to the 20 ms time-to-stretch value) relationships for peak stress are plotted against time-to-stretch shows a similar trend as the example shown in panel B, which is nearly identical for the series of measurements where strain amplitude and strain rate varied within the same trabecula. D) Normalized peak stress data (y-axis and data are identical between panels B and C) are plotted against the reference stress at the start of the ramp-stretch (time = $t_0$), which shows a consistent, roughly-linear relationship between peak stress and reference stress at $t_0$.

passive (small or no nadir/overshoot of the plateau force) for stretches at time points greater than 300 ms after the end of the load clamp. Again, these data show the magnitudes of the detachment and recovery processes scale with the number of crossbridges that remain attached as diastole progresses. Early in diastole (at smaller time-to-stretch values) there is considerable crossbridge detachment and the minimum stress values fall below the reference stretch (= stretch at $t_0$, just before the ramp-stretch began) showing negative values in Fig 3B. As

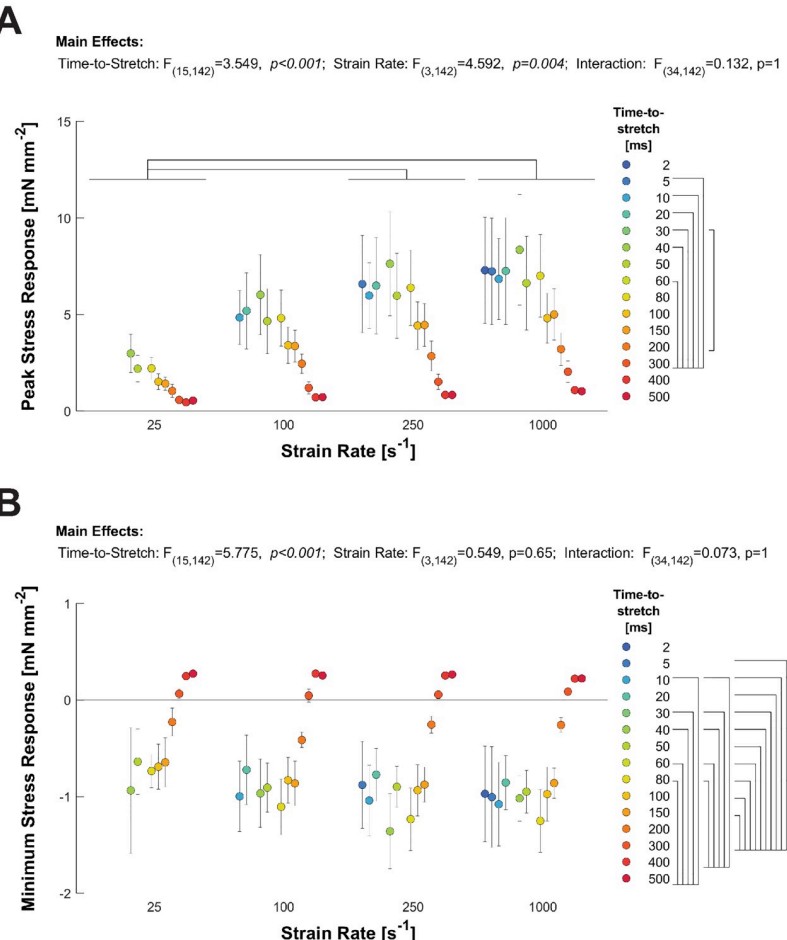

**Fig 3. Effects of strain rate and timing of ramp-stretch on magnitude of the stress-response. A)** Average (±SEM) peak stress values are plotted against time-to-stretch, where each group represents a different strain rate. Peak-stress decreases as time-to-peak increases for each strain rate, consistent with the example data shown in Fig 2. **B)** The average (±SEM) minimum stress values (occurring at $t_2$) are plotted against time-to-stretch, for a series of strain rates. The minimum stress values increase, from slightly negative to slightly positive, as time-to-peak increases. Bars above each graph indicate significant strain rate effects; bars at right of time-to-stretch legend indicate significant time-to-stretch effects.

relaxation progresses, fewer bridges remain bound and the crossbridge detachment processes became less dynamic, appearing more like a passive process (rather than a dip and recovery typical of a stress-response with cycling crossbridges) with minimum stress values approaching the stress value comparable to the diastolic plateau of the reference stress. The lack of strain rate-dependence for the minimum stress-response at $t_2$ was unexpected. The stress-responses of one trabecula at multiple time-to-stretch values and strain rates is shown in Fig 4. Dashed lines are included in the figure to highlight the strain rate dependence of the peak stress-response (red) and the strain rate independence of the nadir (green).

### Effects of strain rate and time-to-stretch on the temporal dynamics of the stress-response

Temporal responses to the ramp stretches also illustrate crossbridge dynamics underlying detachment and force recovery (Fig 5). The half-time between peak stress and minimum stress

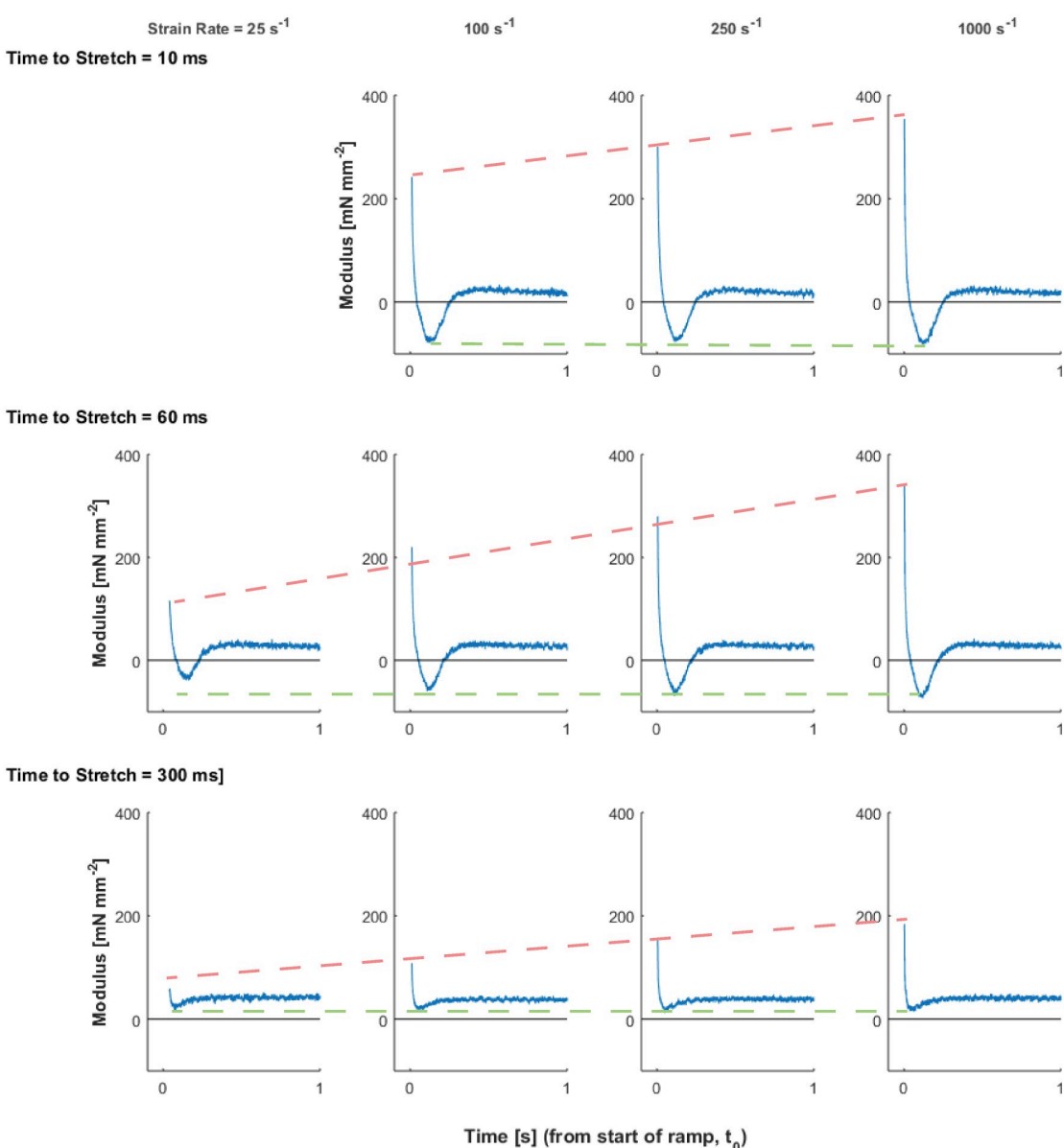

**Fig 4. Example stress-response of one trabecula at three stretch times.** The peak stretch response ($t_0$) shows substantial strain rate dependence. Other parameters do not show significant dependence. Each column shows an equivalent strain rate; each row shows an equivalent time-to-stretch. Red dashed lines highlight the strain rate dependence of the peak stress-response; green dashed lines highlight the relative strain rate independence of the minimum stress-response.

($t_{12}$) decreased as time-to-stretch increased (p = <0.001) and as strain rate increased (p = <0.001, Fig 5A). This half-time was significantly shorter for faster stretches and for stretches occurring 300 ms after the end of the load clamp. The time to minimum stress-response ($t_2$) decreased as time-to-stretch increased (p<0.001; Fig 5B), but was independent of strain rate. The half-time between minimum stress and subsequent maximal stress recovery ($t_{23}$) decreased as time-to-stretch increased (p<0.001; Fig 5C), but was not dependent on strain rate. The independence of the time to minimum ($t_2$) and recovery phase ($t_{23}$) of the stretch response to strain rate is highlighted in the overlaid traces shown in Fig 6.

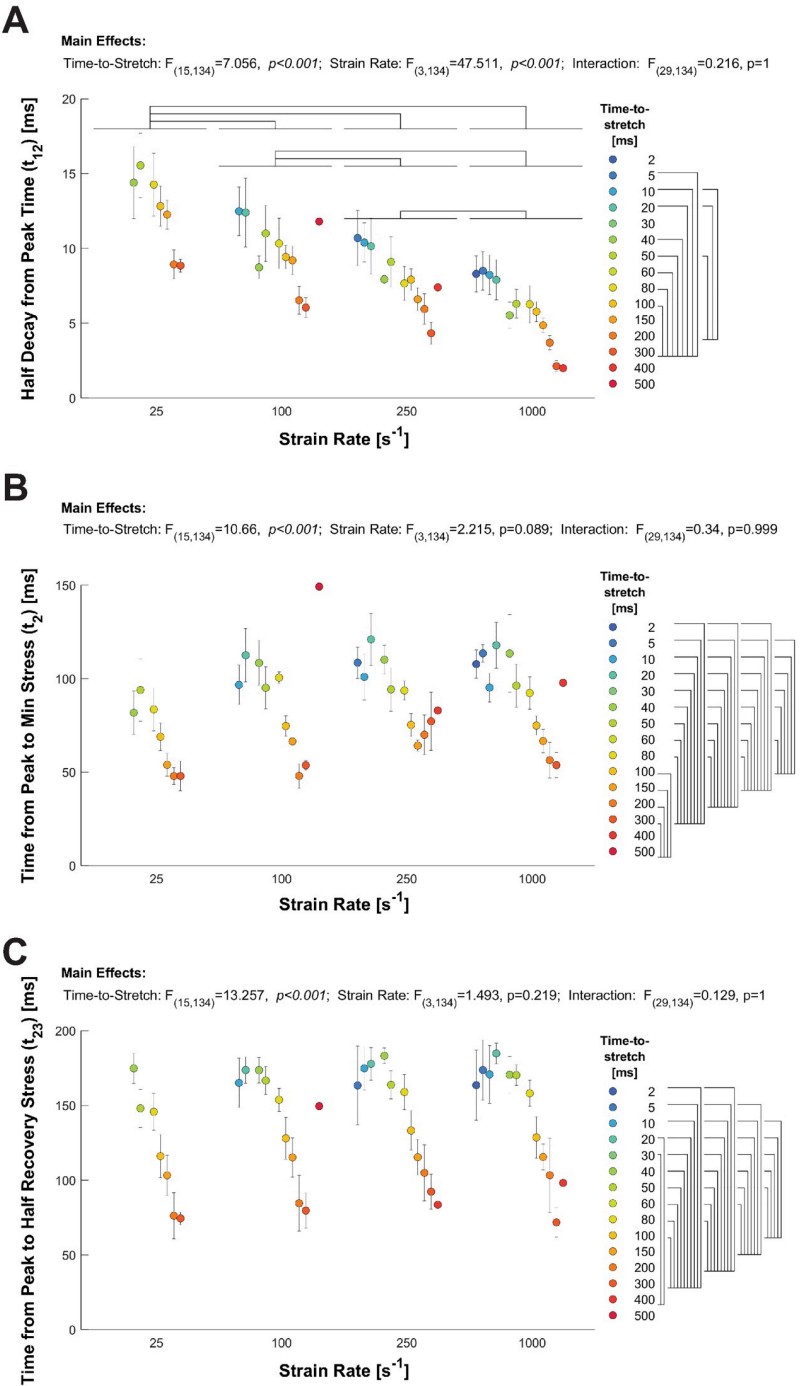

**Fig 5. Effects of strain rate and timing of ramp-stretch on temporal dynamics of the stress-response.** A) The average (±SEM) values for $t_{12}$ (half-time of stress decay) are plotted against the time-to-stretch, following the load clamp, for a series of strain rates. $t_{12}$ decreased as time-to-peak increases for each strain rate, and the average $t_{12}$ value decreased as strain rate increased among all groups. B) The average (±SEM) values for $t_2$ (= time of minimum stress) are plotted against time-to-stretch, for a series of strain rates. $t_2$ decreases as time-to-stretch increases, and this relationship was similar among all strain rates. C) The average (±SEM) values for $t_{23}$ (= half-time of stress recovery) are plotted against time-to-stretch, for a series of strain rates. $t_{23}$ decreases as time-to-stretch increased, and this relationship was similar among all strain rates. Bars above each graph indicate significant strain rate effects; bars at right of time-to-stretch legend indicate significant time-to-stretch effects.

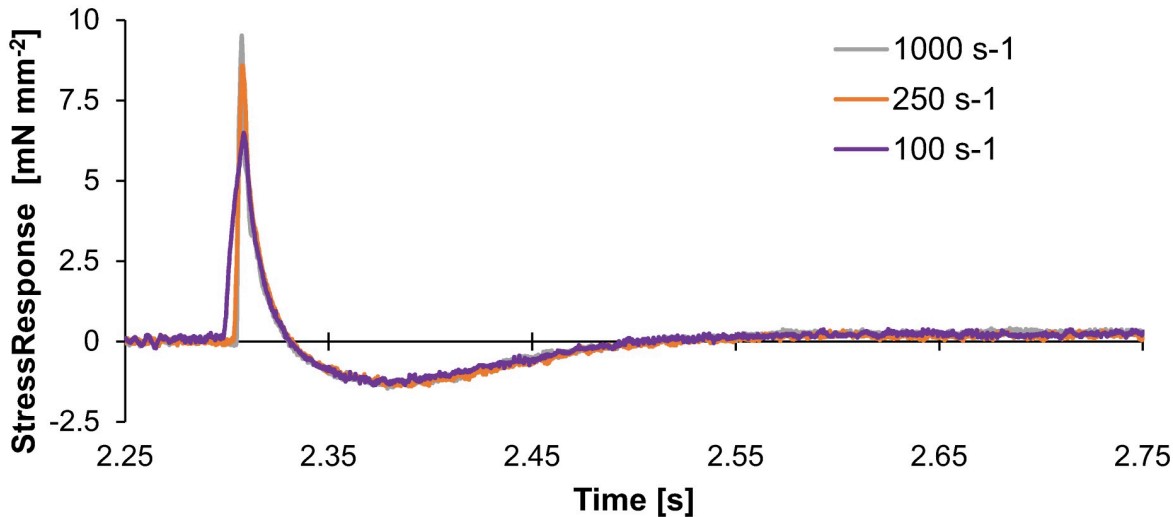

**Fig 6. Stress-response at three different ramp-stretch velocities.** Three stress-responses to a ramp-stretch with increasing strain rate are plotted against time. These are stress-responses from one trabecula at 100, 250, and 1000 $s^{-1}$, 50 ms after the end of the load clamp. Note these traces are slightly shifted in time to align their peaks. These data illustrate that the minimum stress and recovery appear similar for all strain rates, despite the change in peak stress-response. This is consistent with the strain rate independence of the minimum stress-response (Fig 3B) and both the time to minimum stress ($t_2$) and time to stress half recovery time ($t_{23}$) (Fig 5B and 5C).

## Discussion

### Challenges of mechanistic studies of mechanical control of relaxation

Stress and strain are important factors in cardiac function [6], but mechanisms to explain how strain modifies stress are not fully known. When evaluating cardiac (or myocardial) relaxation specifically, it was long held that relaxation could be modified by afterload [7]. Yet, strain in the form of stretch prior to relaxation, also called relaxation-loading, and mechanical dyssynchrony were both highlighted in these analyses as potential confounders. We characterized Mechanical Control of Relaxation as the dependence of the relaxation rate on the end systolic strain rate of a muscle [3]. In contrast, we found that the relaxation-loading (stretch) was not only necessary, but also sufficient, to modify the relaxation rate without changes in afterload. While Mechanical Control of Relaxation was predicted to be related to crossbridge detachment, the mechanisms underlying this phenomenon remained speculative [3, 4]. Modeling performed in these prior studies suggested that the bound crossbridge population was reduced more quickly by stretch, but the experimental protocols did not systematically control for strain rate and time-to-stretch for each trabecula. In this current study we controlled strain rates for each trabecula using a ramp-stretch protocol at increasing time-to-stretch values, following the end of a load-clamp. These data provide novel insights about crossbridge activity as diastole progresses, such as the rapid decline in stress (i.e. the strain rate dependence of $t_{12}$) that indicates crossbridge detachment rate increases with faster strain rates.

### Stress-response to ramp-stretches during relaxation in intact cardiac trabecula

Despite the dynamic changes in stress, some parameters followed predictable features, such as the peak stress-response decreasing as diastole progressed. Relaxation of a twitching cardiac trabecula reflects the natural time course of thin filament deactivation and crossbridge

detachment [8]. The normalized decrease in peak stress correlated with the reference stress just prior to the ramp-stretch during a period of isometric relaxation (Fig 2), indicating that the ramp-stretch distorts bound crossbridges. Among all the data, the magnitude and range of peak stress increased with strain rate (Fig 3A), which suggests that a faster strain rate induces greater distortion of any bound crossbridges. The specific mechanism of this is unclear, but it may follow from greater stress borne by "more distorted" crossbridges due to the faster stretch, which may slow the release of ADP from bound crossbridges and prolong crossbridge binding [9–14].

The half-decay time from the peak to the minimum stress-response ($t_{12}$, Fig 5A) correlates well with prior studies of Mechanical Control of Relaxation where strain rate was dependent on the load-clamp [3, 4]. The decay time becomes shorter with faster strain rates (Fig 5A), which may correlate with a faster relaxation response due in part to faster crossbridge detachment. Surprisingly, the minimum stress and time-to-minimum stress ($t_2$) were not strain rate dependent, although both parameters were dependent on the time-to-stretch (Figs 5B and 6). In general, a larger magnitude nadir was expected with a faster stretch and would likely scale similar to the peak stress-response [15]. In calcium activated permeabilized muscle where the calcium concentration does not change throughout the stretch, the nadir of a stress-response is typically reported to be less than 30 ms [14–17]. In contrast, a relatively long $t_2$ (50–100 ms) was observed in this study. The calcium concentration and thin filament activation status are highly dynamic in intact trabeculae during the relaxation [18, 19], which we observed with the peak stress values decreasing as the time-to-stretch increased (Fig 2). Nonetheless, these data support the mechanism that increased strain rates increase crossbridge detachment rates. This crossbridge detachment would also accelerate thin filament deactivation, potentially attenuating recruitment of crossbridges that would subsequently impact the nadir and recovery of the stress-response.

Later in diastole, the nadir of the stress-response was attenuated, but not eliminated (Fig 4), and the stress-response looks like a passive stress-response of a relaxed muscle. This stress-response would be consistent with very little cytosolic calcium and thin filament activation. Despite the low calcium, the minimum stress-response did retain a small nadir and recovery at most time-to-stretch values suggesting the muscle was not purely passive—and the strain-rate dependent increase in peak stress persisted, even at time-to-stretch values of 300 ms. Therefore, it is possible that small population of crossbridges remain cycling near, or at, end diastole. Resting tone or diastolic crossbridges have been reported in numerous previous studies [20, 21], but it is somewhat surprising that it was present at room temperature.

## Potential impact of time-varying calcium transients

As implied above, the calcium transient and thin filament activation are time varying. For the period that is addressed in this study, i.e. after peak shortening, both the calcium concentration and thin filament activation are declining, which influences the decline in force [3, 8, 22]. While the conditions in the current study differ from traditional studies of myofilament kinetics in permeabilized tissues where the calcium concentration is fixed [6, 9, 14, 15, 17, 20], the substantial strain-rate dependence of the peak stress-response and half-time of force decay from the peak to the minimum stress-response are likely independent of any strain-dependent changes in calcium handling. Length changes of a muscle modify the calcium transient detected over several beats, but such changes take minutes to develop [23–25]. Because this study assesses the stress-response within the same twitch, such long-term changes in calcium flux would not affect the data reported here. During a twitch, rapid length changes, unloaded and loaded shortening, and/or load-clamp protocols result in calcium release from the thin

filament [22, 26–28]. For example, at higher afterloads, cytosolic fluorescent indicators show reduced fluorescence when shortening compared to isometric muscles during early contraction [22, 26]. Because the fluorescent signal is dependent on calcium binding to an indicator, such as aquaporin or Fura-2, it is dependent on the free cytosolic calcium and cannot measure thin filament binding or activation [18, 19]. Thus, it has been reported that changing fluorescence reflects changing thin filament calcium sensitivity [3, 27], as the reuptake of calcium is otherwise unchanged [26]. Cell stretch also increases the rate of transient calcium sparks [29, 30], but sparks do not appear to modify muscle force, unless it induces a contraction. These prior studies suggest that length changes do not substantially influence calcium handling (Ryanodine Receptor (RyR2) or Sarco-endoplasmic reticulum ATPase (SERCA)) in the heart during a twitch.

The time-varying calcium transient in an intact muscle negatively impacts the ability to appropriately model changes in kinetic parameters, because most models assume or describe crossbridge activity during isometric contraction at a static calcium-activation level. Kinetic parameters describing crossbridge detachment and recruitment rates are commonly assessed using step-stretch or sinusoidal length perturbation analysis across multiple frequencies in skinned muscle fibers [6, 9]. It would have been ideal to apply these models of crossbridge distortion and recruitment to the ramp stretch and stress-response measurements from the trabeculae studied here, as these represent similar analysis protocols [6, 15, 17]. However, stress-responses from the intact trabeculae were not well-described by these distortion/recruitment models. Parameter estimates were also inconsistent and varied widely with initial parameter estimates used for different models (unpublished data). Sinusoidal oscillations have been applied to trabeculae, which caused more rapid relaxation, but these oscillations were limited to a single frequency [2]. Alternate sinusoidal analysis protocols were not used in this study because of the rapidly changing stress values during activation and relaxation of the twitching trabeculae before and after the load clamp. Therefore, we anticipate these somewhat simple recruitment and distortion models that typically describe two to four exponential processes associated with crossbridge binding and cycling rates may be insufficient to describe the rates of crossbridge detachment and recruitment during diastole. Perhaps a kinetic model including a time varying calcium flux that modulates crossbridge cycling dynamics is warranted to better describe or fit the contractile responses of intact, twitching cardiac muscle.

## Summary

This study assessed the stress-response of twitching cardiac trabeculae to ramp-stretches that occur during relaxation of the myocardium. These stretches provide better control over the strain rate and timing than typical isotonic load-clamp protocols used to probe Mechanical Control of Relaxation. Time-domain assessment of the peak stress and initial decay of the stress-response revealed expected strain rate-dependencies of cross-bridge distortion and detachment, yet we were surprised that the minimum (nadir) of the stress-response and the dynamics of the stress recovery did not depend upon strain rate. These data support the hypothesis that crossbridge detachment is accelerated by strain rate.

## Acknowledgments

The authors thank Brianna M Schick for data acquisition and curation.

## Author Contributions

**Conceptualization:** Bertrand C. W. Tanner, Charles S. Chung.

**Data curation:** Charles S. Chung.

**Formal analysis:** Bertrand C. W. Tanner, Bradley M. Palmer, Charles S. Chung.

**Funding acquisition:** Bertrand C. W. Tanner, Bradley M. Palmer, Charles S. Chung.

**Investigation:** Bertrand C. W. Tanner, Charles S. Chung.

**Methodology:** Bertrand C. W. Tanner, Bradley M. Palmer, Charles S. Chung.

**Project administration:** Charles S. Chung.

**Resources:** Bertrand C. W. Tanner, Charles S. Chung.

**Software:** Bertrand C. W. Tanner, Bradley M. Palmer, Charles S. Chung.

**Supervision:** Charles S. Chung.

**Validation:** Bertrand C. W. Tanner, Charles S. Chung.

**Visualization:** Charles S. Chung.

**Writing – original draft:** Bertrand C. W. Tanner, Charles S. Chung.

**Writing – review & editing:** Bertrand C. W. Tanner, Bradley M. Palmer, Charles S. Chung.

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
