## [Decision Letter · Decision Letter 0]

15 Nov 2023

PONE-D-23-33513Strain rate of stretch affects crossbridge detachment during relaxation of intact cardiac trabeculaePLOS ONE

Dear Dr. Chung,

Thank you for submitting your manuscript to PLOS ONE. After careful consideration, we feel that it has merit but does not fully meet PLOS ONE’s publication criteria as it currently stands. Therefore, we invite you to submit a revised version of the manuscript that addresses the points raised during the review process.

We look forward to receiving your revised manuscript.

Kind regards,

Daniel M. Johnson, PhD

Academic Editor

PLOS ONE

Journal Requirements:

2. To comply with PLOS ONE submissions requirements, in your Methods section, please provide additional information regarding the experiments involving animals and ensure you have included details on (1) methods of sacrifice, and (2) efforts to alleviate suffering.

   "Funding provided by the National Institutes of Health (R01HL149164 (BCWT), R44HL137603 (BMP), and, R01HL151738 (CSC)), the American Heart Association (23TPA1074093 (BCWT) and 18TPA34170169 (CSC)), and National Science Foundation (2312925 (BCWT) and 1660908 (BMP))." 

Additional Editor Comments:

Both reviewers and myself found this work to be an interesting follow up from previous work from the lab. However, there were a number of issues that need to be resolved, including some mention of the additional limitation brought up by Reviewer 1, and also a number of typographical and presentation errors.

Reviewers' comments:

Reviewer's Responses to Questions

**Comments to the Author**

1. Is the manuscript technically sound, and do the data support the conclusions?

Reviewer #1: Partly

Reviewer #2: Yes

2. Has the statistical analysis been performed appropriately and rigorously? 

Reviewer #1: Yes

Reviewer #2: Yes

3. Have the authors made all data underlying the findings in their manuscript fully available?

Reviewer #1: Yes

Reviewer #2: Yes

4. Is the manuscript presented in an intelligible fashion and written in standard English?

Reviewer #1: Yes

Reviewer #2: Yes

5. Review Comments to the Author

Reviewer #1: The study aims to explore cross bridge mechanisms responsible for the lusitropic effect of small, fast, stretches in the twitching cardiac muscle. The subject may be of interest for both basic and translational research.

I had reviewed a previous version of this manuscript that had been rejected several months ago by a different journal. I must recognize that the changes the Authors have made to the present version of their work recognize most of the limits of the previous version. The changes are in line with the criticisms the Authors had received. The new version is improved even though the Authors should recognize, as a limitation of their work, that the stress vs time data calculated as the difference between the reference and ramp-stretch twitch traces (see Fig 1 central panel) are meaningful only if they assume that the Calcium transient and the myofilament activation level are unaffected by the applied stretches, i.e. the Authors are assuming that the activation and its decay are the same in the reference twitch and in the ramp-stretch twitch. The study provides some advancement in the description of the Mechanical Control of Relaxation reported in a previous paper.

Minor points:

- the Authors should carefully check the text for typos and mistakes (e.g. Fig1 panels: the Authors should pay attention, there is no A B C panels in the Figure as indicated in the legend).

- ‘trabeculae’ is plural in Latin and ‘trabecula’ should be used for the singular case.

Reviewer #2: This is a manuscript which presents an elegant series of measurements on the effect of strain and strain rate on the relaxation of rat cardiac muscle fibers. The work is novel, technically challenging and well presented. It is a follow up to work published by the corresponding author in the J Mol Cell Cardiol in 2017 which established relaxation is dependent upon the rate of stretch just before the start of relaxation. In this study a potential confounding variable was tested (time to stretch and speed of stretch). - The conclusion is that all effects on the time of stretch were compatible with the view that cross bridge detachment and relaxation rate are accelerated by strain rate.

The work is sound and will be of interest to a specialist group of experimentalists.

The conclusions ae sound but not very remarkable.

Minor issues

Introduction Line 13: “relaxaiton”

Methods: subscripts are missing for all chemical formulae e.g. KHCO3 - KHCO3.

Just before Fig 1 legend: “was acquired by repeating the load-clamp was repeated”

Fig 1 is missing A, B, C labels.

Fig 1C It is not clear what t12 refers to. It would be useful to include t0, t1 & t2

There are two Fig 6’s, one is Fig 5

6. PLOS authors have the option to publish the peer review history of their article (what does this mean?). If published, this will include your full peer review and any attached files.

Reviewer #1: No

Reviewer #2: No

---

## [Author Response · Author response to Decision Letter 0]

15 Dec 2023

Review Comments to the Author

We thank the reviewers for their comments regarding the manuscript. In addition to Editor requested items (Funding, Data), we have worked to correct typos, minor errors, and inconsistent labeling, and we have the discussion to enhance clarity.

Reviewer #1: The study aims to explore cross bridge mechanisms responsible for the lusitropic effect of small, fast, stretches in the twitching cardiac muscle. The subject may be of interest for both basic and translational research.

I had reviewed a previous version of this manuscript that had been rejected several months ago by a different journal. I must recognize that the changes the Authors have made to the present version of their work recognize most of the limits of the previous version. The changes are in line with the criticisms the Authors had received. The new version is improved even though the Authors should recognize, as a limitation of their work, that the stress vs time data calculated as the difference between the reference and ramp-stretch twitch traces (see Fig 1 central panel) are meaningful only if they assume that the Calcium transient and the myofilament activation level are unaffected by the applied stretches, i.e. the Authors are assuming that the activation and its decay are the same in the reference twitch and in the ramp-stretch twitch. The study provides some advancement in the description of the Mechanical Control of Relaxation reported in a previous paper.

RESPONSE

We thank the reviewer for their productive and positive comments. We have now expanded our discussion regarding the importance of time-varying calcium in a section called “Potential Impact of Time-varying Calcium Transients” at the end of the discussion. This section more carefully assesses how length changes might modify the calcium transient and modeling of the data. Specifically, multiple prior reports suggest that the impact within a twitch is minimal, whereas most changes would take minutes to develop and would not show up within the twitches that were analyzed in this study.

Minor points:

- the Authors should carefully check the text for typos and mistakes (e.g. Fig1 panels: the Authors should pay attention, there is no A B C panels in the Figure as indicated in the legend).

- ‘trabeculae’ is plural in Latin and ‘trabecula’ should be used for the singular case.

RESPONSE

We thank the reviewer for the careful review and comments. We have checked for errors and have carefully considered the singular or plural use of trabecula. Figure 1 has been corrected and labels in 1C added; we apologize for not catching this during the conversion of the figure for submission.

Reviewer #2: This is a manuscript which presents an elegant series of measurements on the effect of strain and strain rate on the relaxation of rat cardiac muscle fibers. The work is novel, technically challenging and well presented. It is a follow up to work published by the corresponding author in the J Mol Cell Cardiol in 2017 which established relaxation is dependent upon the rate of stretch just before the start of relaxation. In this study a potential confounding variable was tested (time to stretch and speed of stretch). - The conclusion is that all effects on the time of stretch were compatible with the view that cross bridge detachment and relaxation rate are accelerated by strain rate.

The work is sound and will be of interest to a specialist group of experimentalists.

The conclusions are sound but not very remarkable.

Minor issues

Introduction Line 13: “relaxaiton”

Methods: subscripts are missing for all chemical formulae e.g. KHCO3 - KHCO3.

Just before Fig 1 legend: “was acquired by repeating the load-clamp was repeated”

Fig 1 is missing A, B, C labels.

Fig 1C It is not clear what t12 refers to. It would be useful to include t0, t1 & t2

There are two Fig 6’s, one is Fig 5

RESPONSE

We thank the reviewer for the careful assessment. We have corrected the errors in the text and revised Figure 1 to include labels that were lost when converting to the format for submission.

---

## [Editor Report · Decision Letter 1]

2 Jan 2024

Strain rate of stretch affects crossbridge detachment during relaxation of intact cardiac trabeculae

PONE-D-23-33513R1

Dear Dr. Chung,

We’re pleased to inform you that your manuscript has been judged scientifically suitable for publication and will be formally accepted for publication once it meets all outstanding technical requirements.

Kind regards,

Daniel M. Johnson, PhD

Academic Editor

PLOS ONE

Additional Editor Comments (optional):

Please check the proofs of the paper carefully, as there are still spelling errors which need to be eliminated. For example, ''Ryanode' should be 'Ryanodine' in the last part of the new section added.
---

## [Editor Report · Acceptance letter]

23 Feb 2024

PONE-D-23-33513R1 

PLOS ONE

Dear Dr. Chung, 

I'm pleased to inform you that your manuscript has been deemed suitable for publication in PLOS ONE. Congratulations! Your manuscript is now being handed over to our production team.

Kind regards, 

on behalf of

Dr. Daniel M. Johnson 

Academic Editor

PLOS ONE